# Effect of High-Hydrostatic-Pressure Processing and Storage Temperature on Sliced Iberian Dry-Cured Sausage (“*Salchichón*”) from Pigs Reared in Montanera System

**DOI:** 10.3390/foods11091338

**Published:** 2022-05-04

**Authors:** Rosario Ramírez, Antonia Trejo, Jonathan Delgado-Adámez, María Jesús Martín-Mateos, Jesús García-Parra

**Affiliations:** Technological Agri-Food Institute (INTAEX), Center for Scientific and Technological Research of Extremadura (CICYTEX), Avda, Adolfo Suárez s/n, 06071 Badajoz, Spain; mariarosario.ramirez@juntaex.es (R.R.); tonitrejoal@gmail.com (A.T.); jonathan.delgado@juntaex.es (J.D.-A.); mariajesus.martinma@juntaex.es (M.J.M.-M.)

**Keywords:** dry-cured fermented sausage, *salchichón*, *Montanera*, refrigerated storage, high-pressure processing, lipid oxidation, sensory analysis

## Abstract

The top-quality “*salchichón*” (a fermented dry-cured sausage) is manufactured from Iberian pigs reared outdoors. This work aims to evaluate the effect of hydrostatic high pressure (HHP) and storage temperature on sliced vacuum-packaged top-quality Iberian “*salchichón*”. Two types of “*salchichón*” (S1 and S2, manufactured at different companies) were processed at 600 MPa for 8 min and stored at 4 and 20 °C for 180 days. Microbiological, physicochemical, and sensory changes were evaluated. Microbiological counts were reduced by HHP treatment and also generally decreased during storage at both temperatures. Lightness and redness of slices decreased during storage at 20 °C, while yellowness values increased. Changes in color were also observed in sensory analysis of the dry-cured sausages. HHP increased lipid and protein oxidation values in S1, whereas protein oxidation increased at 20 °C in S2. S1 was more affected by HHP while S2 was more affected by the temperature of storage. Therefore, despite both products belonging to the same commercial category, slight differences in the composition of both products and/or differences in packaging determined a different behavior after HHP treatment and during storage at different temperatures.

## 1. Introduction

Iberian pig is an autochthonous swine breed from the southwestern Iberian Peninsula. This breed is traditionally reared on a Mediterranean ecosystem, called “*Dehesa*”, composed mostly of evergreen oaks (*Quercus ilex* and *Quercus rotundifolia*). The feeding system of pigs reared extensively is almost exclusively based on grass and acorns (called *Montanera*) for at least two months before slaughter, which gives high stability to Iberian meat products from pigs reared under these conditions [1], but only top-quality dry-cured Iberian meat products are manufactured from pigs reared in extensive conditions, since most are reared in semi-intensive conditions and fed with commercial feeds. The feeding system determines the stability of sliced Iberian dry-cured meat products during storage [2].

Iberian “*salchichón*” is a traditional dry-cured sausage prepared with meat and minced fat from Iberian pig, seasoned with a mixture of spices and starter cultures (optional). Fermentation is a decisive phase of the “*salchichón*” ripening process, due to physical, biochemical, and microbiological changes produced during this period [3]. In 2020, the consumption of meat products in Spain (fermented, dry-cured, and cooked) increased compared to the previous year. The “*salchichón*” consumption represented the 10% of the total [4].

Nowadays, the sliced format has become more popular instead of the whole piece, due to changes in consumer habits [5]. However, slicing, packaging, or other handling of the product involves certain risks that could negatively affect the quality of the product, such as microbiological contamination [6] or greater exposures to light and oxygen, factors that affect the stability of “*salchichón*”.

“*Salchichón*” is generally considered a microbiologically safe product. However, processes such as the slicing or packaging can compromise their safety. Packaged sliced dry-cured sausages require a long shelf-life, and they are normally stored at refrigeration temperatures. Producers also demand a possible storage without refrigeration of the sliced product, which presents a challenge for the Iberian meat products sector. Microbiological spoilage and lipid oxidation reactions in food are affected by temperature; both are enhanced, with increasing temperature affecting final quality. Therefore, shelf-life of “*salchichón*” is influenced, among other factors, by both the storage temperature and the presentation format [7,8,9].

High-hydrostatic-pressure (HHP) technology is a post-process decontamination control technology for packaged products, especially in those foods whose functional, nutritional, or sensory characteristics can be affected by thermal treatment [7]. Industries are applying HHP treatments to increase the safety of these products due to prepackaging manipulation. Sometimes, some undesirable-quality damages may appear. This is especially important for top-quality Iberian meat products (from 100% Iberian pigs reared outdoors). High-pressure-treated products are normally stored in refrigeration conditions; however, recently it is being explored whether the application of HHP could increase the safety of dry-cured meat products even when they are stored at room temperature.

Application of HHP and the effect of storage temperature of processed products have been studied in different Iberian dry-cured meat products [7,9,10], but the information about the effect on fermented products such as “*salchichón*” is limited. Moreover, intrinsic differences caused by the composition of the fermented product (salt, fatty acids composition, water activity, microbiological profile, etc.) could modify their stability after HHP and during the subsequent storage. No previous study has evaluated the effect of HHP and storage conditions on sliced top-quality “*salchichón*”. In addition, two types of products manufactured in different companies were studied to obtain more representative results. Consequently, the aim of this work was to evaluate post-packaging changes associated with hydrostatic high pressure and the temperature of storage on two types of sliced Iberian “*salchichón*” with different composition.

## 2. Materials and Methods

### 2.1. Material

Iberian “*salchichón*” was manufactured in two independent local industries in the southwest of Spain, according to their commercial specifications. Both products (“*salchichón* S1 and S2) could be considered as the same top-quality commercial category Iberian “*salchichón*” (Black category, according to the RD4/2014 for ham, shoulder, and loin) [11]. Specifications of assay products are shown in Table 1. Pigs were fed on acorns and pasture while being reared outdoors in Mediterranean evergreen forests. Both lean and fat used for the manufacture of “*salchichón*” were from 100% Iberian breed. “*Salchichón*” were manufactured with minced Iberian lean meat and fat, mixed with salt, spices, and authorized additives, macerated, stuffed in a 60 to 65 mm diameter sausage natural casing, and subjected to a suitable curing process for at least 120 days. During maturation, temperature and relative humidity (RH) in the refrigeration chamber must be progressively increased and decreased, respectively. Initially “*salchichón*” was matured at 5 to 7 °C and 75 to 85% RH to finally reach 15 to 20 °C/55 to 65% RH at the end of the maturation process. The approximate weight of each piece was 1 kg. Dry-cured sausages from the same production batch were sliced and vacuum-packaged (100 g each package). A total of 120 packages were bought from each industry (100 packages were used for the experiment, five packages were used for the characterization of the product, and 15 packages were used for the training of the panelists in the sensory analysis).

### 2.2. Experimental Design

Two different treatments were performed in each type of “*salchichón*”: control packages (sliced vacuum packages not subjected to any treatments) and HHP (sliced vacuum packages subjected to a high-pressure treatment) and were stored at two different temperatures (4 and 20 °C). Packages were stored for 180 days and they were sampled at three different times: at the beginning of storage (day 0), at day 90, and at 180 days. Storage time was decided according to the date of preference consumption of these products, which is generally less than 6–8 months. Five packages per group were analyzed for composition analysis (*n* = 5). A total of 100 packages of sliced “*salchichón*” were analyzed. For the microbiological and physical–chemical analysis, 50 sliced sausage packages were utilized, whereas another 50 packages were used for sensory analysis.

### 2.3. High-Pressure Processing Conditions

Packages were pressurized at 600 MPa for 8 min in a semi-industrial hydrostatic pressure unit with 55 L of capacity (Hiperbaric Wave 6000/55; Burgos, Spain). The initial temperature of the water inside the vessel was 10 °C. Time to reach 600 MPa was about 4 min. Decompression of the vessel was instantaneous (<3 s). When high-pressure treatments at 600 MPa were applied starting at 10 °C, the temperature reached in the sample was approximately 25 °C due to the adiabatic compression, increasing due to adiabatic heating approximately 2.5 °C/100 MPa [12]. These are the optimal conditions to inactivate *Listeria monocytogenes* in “*salchichón*” [8]. Packages were stored in darkness at refrigeration (4 °C) and in a separated room at 18–22 °C.

### 2.4. Initial Characterization (Water Activity, pH, Protein, Moisture, and Fat Content)

For composition analysis, five different packages of sliced “*salchichón*” were analyzed at the first stage of storage. pH was measured by a pH-meter Crison pH 25 + (Crison, Barcelona, Spain). The moisture content was calculated by standard procedure, which involves a weight loss by drying the sample in an drying oven at 104 °C until a constant weight is reached. Water activity (a_w_) was measured by means of a Labmaster-aw meter (Novasina AG, Lachen, Switzerland). Protein content was determined by Kjeldahl method and fat content was determined gravimetrically by extraction with chloroform:methanol (2:1).

### 2.5. Fatty Acid Profile

Fatty acid methylesters (FAMES) were analyzed using an Agilent 6890 gas chromatograph (Agilent Technologies, Santa Clara, CA, USA), equipped with a flame ionization detector (FID) following the methodology of Trejo et al. [13]. Results are expressed as a percentage of total fatty acid methylesters.

### 2.6. Microbiological Analysis

Ten grams of each sample was aseptically taken and homogenized with 90 mL of peptone water (Merck, 1.07043) in a laboratory blender (Stomacher^®^ 400 Circulator; Seward, Worthing, UK). Serial decimal dilutions were made in sterile peptone water, and 1 mL samples of appropriate dilutions were poured or spread onto total count and selective agar plates. Mesophilic aerobic counts, lactic acid bacteria (LAB), *Staphylococcus aureus*, *Clostridia perfringens*, molds and yeasts, total coliforms, *E. coli*, *Salmonella* spp*.,* and *L. monocytogenes* analyses were performed according to the methodology described in Amaro-Blanco et al. [5]. Results were expressed as log_10_ CFU (colony forming units) g^−1^. Analyses were performed in duplicate.

### 2.7. Instrumental Color

Instrumental color determinations were carried out with a Minolta CM-5 spectrophotometer (Konica Minolta, Osaka, Japan), using an illuminant D65, a 10⁰ standard observer, and measuring area of 30 mm. The color parameters L* (lightness), a* (redness), and b* (yellowness) in the CIELAB color space were evaluated.

### 2.8. Lipid Oxidation

Lipid oxidation was assessed by thiobarbituric acid reactive substances (TBA-RS) [14], using 4 g of sample, homogenized with perchloric acid (3.86%) and BHT (4.2%) at 11,500 rpm for 45 s with ice cooling, then centrifuged, and supernatant filtered. TBA-RS values were calculated from a standard curve of tetraethoxypropane (TEP) and expressed as mg malondialdehyde (MDA) kg^−1^ meat (mg MDA kg^−1^).

### 2.9. Protein Oxidation

Protein oxidation was evaluated by measuring carbonyl groups formed during incubation with 2,4-dinitrophenylhydrazine (DNPH) in 2N HCl following the method described by Oliver et al. [15] and expressed as nmol carbonyls mg^−1^ protein.

### 2.10. Sensory Analysis

The tasting panel was composed of eight trained judges. Two slices from each batch were presented to each panelist. The following descriptors were evaluated: lean and fat color, odor intensity, unpleasant odors, hardness, juiciness, saltiness, acidity, sweetness, flavor intensity, cured aroma, and rancidity. The methodology is detailed in Trejo et al. [13]. Ethical review and approval were waived for this study due to sensory analysis of commercial foods does not require the approval of an ethics committee as there is no intervention in humans.

### 2.11. Statistical Analysis

Five samples per treatment were analyzed (*n* = 5). Two-way analysis of variance (ANOVA) of the effect of treatment applied (high-pressure treatment, storage temperature) was performed using SPSS, Version 21.0 (SPSS Inc., Chicago, IL, USA). A one-way ANOVA was applied to evaluate the changes during storage. When ANOVA showed significant differences, Tukey’s HSD test was applied to compare the mean values. Another one-way analysis of variance (ANOVA) was applied to evaluate differences between both types of “*salchichón*”. Mean values with standard deviation are reported. The relationship between parameters was assessed by the calculation of principal component analysis (PCA).

## 3. Results and Discussion

### 3.1. Initial Characterization

Physical–chemical composition (Table 2) was rather different between both products. S1 presented higher pH, a_w_, %moisture, and %protein content than S2, whereas fat content was similar in both products. The final pH of fermented meat products is a consequence of the development of LAB during the maturation process. LAB produce lactic acid during maturation process, which decreases pH and inhibits microbial growth of other species. A common practice in some meat industries to achieve a pH suitable for the maturation process consists of adding commercial starters, which seems to be the cause of the lower pH in S2 than in S1. Product S2 has a lower moisture content, which ensures the safety of the product. S2 would have been manufactured following a drying process more intense with higher weight loss during maturation than S1. In fact, S2 products had smaller diameter than S1, which also involves a greater loss of moisture during the ripening process. Values of pH and a_w_ agree with those reported by Cava et al. [8] and Martín et al. [9] for other Iberian fermented dry-cured products such as “*salchichón*” or “*chorizo*”, respectively, although values of S1 could be considered to be high. Fat content is considered within the appropriate range for dry-cured meat products [8].

Fatty acids composition (Table 3) was different in S1 and S2, and only minor fatty acids, such as C12:0, C17:0, C20:0 and C20:1, did not show significant differences. Levels of saturated (C14:0, C16:0, C18:0) and some monounsaturated fatty acids (C16:1, C17:1) were higher in S1 than S2, but S2 presented higher levels of C18:1 and polyunsaturated fatty acids (C18:2 and C18:3). Fatty acids profile of pork is affected by the feeding systems of animals. Both types of “*salchichón*” had high levels of oleic acid, which is a feature of dry-cured meat products from pigs reared in *Montanera*. Tissues of pigs reflect the fatty acid composition of the feeds consumed during fattening period [16]. Differences in fatty acids profile could be attributed to the different meat cuts utilized for the manufacture of the “*salchichón*”, or to the differences in the feeding composition. In this respect, Tejerina et al. [17] reported important differences in the composition of acorns and pasture, the main feeding of pigs in *Montanera*. Fatty acids profile could affect the stability of dry-cured meat products after processing or storage, since unsaturated fatty acids are easily oxidized [18].

Some factors have to be taken into account when applying HHP to a food system, such as fatty acid profile or salt/moisture/fat content. Several studies have highlighted that there is a reduction of HHP efficiency due to the high presence of solutes and some salts (NaCl, KCl) in addition to low a_w_ [19]. Moreover, these physicochemical properties and storage conditions serve as additional obstacles to pathogen growth and have been shown to inhibit the recovery of injured cells due to HHP application during prolonged storage of a product [20,21].

### 3.2. Effect of HHP and Storage on the Microbiology

HHP produced an inhibitory effect (*p* < 0.05) on mesophilic aerobic counts (Table 4). The effect could be observed after processing or during storage, since the treatment could also produce sublethal changes, which are evident not just after processing [22]. In addition, storage at 20 °C also reduced the counts of mesophilic aerobic counts in S1 and S2 in comparison to refrigeration temperatures. Initial counts were similar in S1 and S2, although the effect of processing and storage was more intense in S2 than S1. In fact, reductions after processing and storage were higher in S2 than S1.

LAB counts were also significantly affected by HHP, although reductions were not significant immediately after processing. Similarly to mesophilics, reductions were more marked in S2 than S1. Storage at 20 °C also reduced the LAB counts. Generally, in both products (S1 and S2), the counts for *S. aureus* and *Cl. perfringens* were below the detection limit at the end of the storage. *S. aureus* were higher in S1 than S2, and they were reduced by HHP. Molds and yeasts were notably inactivated by the HHP (*p* < 0.05) in S2, and recovery of surviving cells was not observed during storage. A similar trend was observed in S1, although without significant differences. In HHP-treated “*salchichón*”, the mesophilic aerobic counts, LAB, *S. aureus*, *Cl. perfringens*, and molds and yeast counts slowly decreased until the end of the storage period (day 180). Similar results for the microorganism’s reduction pattern were reported by Rubio et al. [23] in vacuum-packaged “*salchichón*” treated at 500 MPa for 5 min and stored at 4 °C for 210 days.

Coliforms counts were decreased by HHP (day 0 in S1; day 0 and 90 in S2). The storage temperature (day 90 and/or 180) also showed a significant effect with higher counts at 4 °C than at 20 °C. *E. coli* counts were below the detection limit, except in S1 at day 0. The presence of pathogens such as *L. monocytogenes* and *Salmonella* spp. (data not shown) was also evaluated. Analysis showed an absence (in 25 g) at the beginning of storage in S1 and S2. HHP was effective for the inactivation of pathogenic microorganisms, such as *Salmonella* in fermented sausages at 400 MPa/10 min/17 °C [24] and *L. monocytogenes* in sliced vacuum-packed chorizo treated at 600 MPa/10 min/18 °C [25].

Despite that those commercial starters were used just for the manufacture of S2, both products presented similar initial counts of LAB. In addition, levels of mesophilics aerobic counts and mold and yeast were similar. However, the levels of *S. aureus*, coliforms, and *E. coli* were higher in S1. The inclusion of a commercial starter, which effectively decreased the pH during maturation, limited the growth of pathogens in S2 compared to S1. Final pH is a key factor to maintain the safety of fermented meat products. In addition, final pH was also decisive for the effectiveness of the processing, since significant differences between S1 and S2 were observed: inactivation of mesophilic aerobic counts, LAB, and total coliform counts after HHP and during storage, with the inactivation of microorganisms higher in S2 than in S1.

The differences in the effects, caused by the HHP treatment, regarding the microorganisms’ inactivation are due to diverse factors. Changes in the composition of meat products produced a differential effect of processing [26,27]. In general, the higher the moisture content in the product, the greater the effectiveness of HHP, since pressure is water-transmitted. However, differences in moisture or a_w_ would not explain our results, since both were higher in S1 than S2. However, the lower pH of S2 than S1 could have favored the inactivation of microorganisms during processing and the subsequent storage, since acid conditions limit the recovery of microorganisms. This factor could explain the differential effect of processing and the pattern during storage of S1 and S2. As a result, differences in the levels of salt, additives, or spices could also modify the effect of HHP-processing.

The increase of total coliforms after 90 days of storage could be caused by the displacement of the populations of mesophiles, lactic acid bacteria, etc., (less competition for nutrients, space, etc.) that occur mainly as a consequence of HHP and, to a lesser extent, vacuum packaging. This fact is more evident in storage at a temperature of 20 °C, due to the closeness to the optimum growth temperature for this group of microorganisms. After the increase during the first 90 days of storage, a decrease was observed in the counts of total coliforms, as well as in the rest of the microorganisms, due to the lack of nutrients and unfavorable storage conditions for them. Therefore after 180 days, HHP and storage at 20 °C reduced the level of microorganisms’ counts, even coliforms count. This could increase the safety of the product in vacuum-packaging conditions. In line with our results, Cava et al. [8] studied the effects of HHP and storage temperature in dry-cured Iberian “*salchichón*” packaged in thick slices. They found that HHP initially reduced the mesophilic aerobic counts and yeast and molds counts during storage; however, the storage at room temperature decreased the level of these counts compared to storage at 4 °C. In addition, Cava et al. [7,8] reported that storage temperature at 18 °C caused a significantly greater detrimental effect on *L. monocytogenes* population than colder storage (4 °C) in Iberian dry-cured chorizo and “*salchichón*”. Moreover, Serra-Castelló et al. [28] described that storage at room temperatures (25 °C) resulted in an certain inactivation (1 log reduction) of *L. monocytogenes* in experimentally inoculated traditional dry-cured ham.

### 3.3. Effect of HHP and Storage on the Instrumental Color and Lipid and Protein Oxidation

Instrumental color changes of the two types of dry-cured sausages (S1 and S2) are presented in Table 5. In S1, HHP decreased the lightness (CIE L*) of slices at day 0; however, at days 90 and 180, the effect of the treatment did not modify this parameter. At day 180, the storage temperature at 20 °C reduced the values of CIE L* compared with those stored at 4 °C, which showed values similar to day 0. In S2, HHP did not modify CIE L*. At day 90, slices stored at 20 °C showed higher values of CIE L* than those stored at 4 °C, although at day 180 an opposite trend was found, and slices stored at 20 °C had lower CIE L* values than those stored at 4 °C. In addition, during storage at 20 °C of S2, both control and HHP slices showed a significant decrease in lightness. Storage at room temperature promotes the development of darker color in sliced products compared to refrigerated storage, as our results show a decrease in CIE L* regardless of the application of HHP. Other authors [7] found this effect in Iberian dry-cured chorizo treated by HHP during storage at room temperatures.

In both S1 and S2, redness (CIE a*) was not modified after HHP at refrigerated storage. However, storage at room temperature in S1 showed significant differences: redness values at day 90 showed significantly higher values than at days 0 and 180. In S2, only slices stored at 20 °C showed a significant decrease in CIE a* during storage.

In general, the color of other Iberian dry-cured sausages, such as chorizo, was not modified after HHP [7,9]. However, the final color of chorizo is mostly reached with the addition of red paprika; meanwhile, this spice is not used for the manufacture of “*salchichón*”, and its red color is mostly attributed to the formation of nitrosylmyoglobin. Reductions in redness could be explained in dry-cured meat products by the denaturation of nitrosylmyoglobin, which could be promoted at room temperature [29].

Yellowness (CIE b*) at refrigerated storage in control samples of both S1 and S2 did not show significant differences. In addition, yellowness in S2 samples was not modified after HHP treatment at the same temperature. In S1, HHP samples at day 180 saw a significant increase in CIE b* value. Regarding storage at room temperature, S1 control samples did not modify significantly, while after HHP treatment, their yellowness increased after 90 and 180 days. In the case of S2, the increase in storage time significantly increased CIE b* value. Generally, increases in yellowness are associated with a higher lipid oxidation and rancidness perception [30], which could be negative for the preservation of the characteristics of Iberian “*salchichón*”. In contrast to the effect of HHP in “*salchichón*”, in sliced Iberian chorizo, CIE b* was not affected by HHP [7,13].

S1 and S2 had similar lightness but different redness (S2 > S1) and yellowness (S2 > S1). The effects of processing and storage, although they showed similar patterns, were more intense in one product than in the other. Concretely, effect of HHP on yellowness after the treatment was more marked in S1 than S2. This fact could be associated with the higher oxygen permeability of packages of S1 than S2, which could favor the oxidation reactions after processing and especially during storage of treated products, despite S1 presenting lower polyunsaturated fatty acids content than S2. The use of plastic bags with lower oxygen permeability could increase stability of processed products of S1 during storage.

Storage of sliced “*salchichón*” at 20 °C decreased CIE L* in both types of “*salchichón*”, decreased CIE a* in S1, and increased CIE b* in S2 (day 90). In line with these results, Cava et al. [8] found that color changes of “*salchichón*” were more affected by temperature of storage than by HHP. Storage at room temperature reduced lightness and increased yellowness of “*salchichón*”. In other Iberian dry-cured sausages, such as chorizo, color parameters were only modified after HHP at the beginning of storage [9]. The color of “*salchichón*” could be more instable than chorizo, since the latter is manufactured with red paprika, which provides the characteristic red color to meat products. Modifications of color parameters at room temperature could be associated with further development of oxidative reactions at 20 °C than at 4 °C, which could significantly affect the original color of “*salchichón*” by changes in the nitrosylmyoglobin [29] and the chemical structure of fat, such as lipid oxidations.

In S1, TBA-RS values (Table 6) were affected by HHP and storage temperature: interaction between both factors was significant at day 90 and 180. Application of HHP increased TBA-RS values at day 90 and 180. Regarding effect of temperature on lipid oxidation, TBA-RS values of HHP-treated packages of S1 at 20 °C were lower than values at 4 °C. Changes in lipid oxidation in S1 after HHP are in line with increases of CIE b*. On the other hand, in S2, TBA-RS values of HHP-treated slices at 4 and 20 °C significantly increased during the storage, whereas the control remained unchanged at the same storage temperatures. The results of S1 and S2 regarding the temperature of storage are unexpected. However, Martin et al. [9] and Trejo et al. [13] found reductions of TBA-RS values in a traditional chorizo stored at room temperature. Increased temperature has a promoting effect on the secondary oxidation of lipids and accelerates the rates of formation and decomposition of hydroperoxides simultaneously, this effect being stronger on hydroperoxides decomposition than on their generation [31]. Decreases in TBA-RS values during storage are not common, but this could be attributed to the reaction of MDA with amino acids, sugars, and other compounds in the formulation [32]. It is possible that study of lipid-derived volatile formation could clarify these results, since TBA-RS and hexanal content did not follow the same pattern in previous studies in “*salchichón*” [8]. Protein oxidation in S1 was not affected by processing or by the storage temperature at days 0 and 90. However, at day 180, protein oxidation was higher in HHP-treated slices than the control. During storage, HHP-treated slices at 4 °C showed a significant increase in the levels of protein oxidation. On the other hand, in S2, protein oxidation was not affected by processing or by the storage temperature at days 0 and 90, similar to results for S1. At day 180, slices stored at 20 °C showed higher development of protein oxidation than those stored at 4 °C. During storage, all groups of S2 increased the levels of protein oxidation.

Cava et al. [8] reported a different effect of HHP and the storage temperature in dry-cured “*salchichón*”. They reported increases of TBA-RS and reductions in protein oxidation after HHP (600 MPa/8 min), as the storage at room temperature increased lipids and protein oxidation development. Differences in the results show the complexity of these reactions, and they could be explained by (i) the different presentations of products (thick slices of 2 cm vs. thin slices of 2–3 mm); (ii) different plastic packaging characteristics (i.e., oxygen permeability), (iii) the longer time of storage in the current study than previous studies, (iv) differences in the manufacture process (mincing grade, additives, maturation conditions, etc.), (v) quality characteristics of raw material (in this case, “*salchichón*” was prepared from Iberian pigs reared outdoors), etc.

With respect to the effect of HHP on protein oxidation development, there is not an agreement about the effect on dry-cured meat products, since occasionally processing did not increase protein oxidation development after HHP or during storage in sliced dry-cured shoulder from pigs reared in *Montanera* [5]. Rubio et al. [23] did not find a prooxidant effect of high-pressure treatment (500 MPa/5 min) on sliced “*salchichón*” during refrigerated storage (6 °C/210 days). However, Fuentes et al. [33] found increases in lipid oxidation (hexanal content) and protein oxidation values as a result of HHP treatment (600 MPa/6 min/12 °C) of sliced Iberian dry-cured hams during storage (30 days). Similarly, Cava et al. [7] reported an increased susceptibility of HHP-treated Iberian chorizo to suffer lipid oxidation during storage, as well as higher formation of carbonyls compounds after 120 days of storage in HHP-treated Iberian chorizo and in products stored at room temperature compared to refrigerated storage. This agrees with the higher protein oxidation at day 180 in HHP-treated slices of S1 and the higher development of protein oxidation at 20 °C than at 4 °C in S2.

Regarding differences between both types of “*salchichón*” (PS1–S2), initially, S2 presented higher values of lipid and protein oxidation than S1, and during storage, values significantly increased in S2 whereas they remained unchanged or decreased in S1. Probably, the higher polyunsaturated fatty acids of S2, such as linoleic and linolenic acids, would have enhanced this reaction in sliced dry-cured meat products, since they are oxidized faster [34]. The mechanisms behind protein and lipid oxidation are not the same, but they are interconnected because both processes may be influenced by similar prooxidant and antioxidant factors [35]. Fuentes reported that pre-sliced dry-cured ham was more susceptible to oxidative reactions after HHP and subsequent refrigerated storage, since paired or cross-linked reactions between lipid and protein oxidation may have occurred. However, the literature provides conflicting results, as positive correlations between protein and lipid oxidation are not always found. In fact, the protein oxidation after HHP is a very recent topic [36]. Processing by HHP could make “*salchichón*” more sensitive to suffer oxidative reactions during storage, which are enhanced at temperatures higher than refrigeration. In addition, initial lipid and protein oxidation was higher in S2, probably due to a more developed maturation process in S2. In fact, moisture and a_w_ were lower in S2 than S1. Other factors, such as the oxygen permeability of packaging, would also affect the stability of this product during storage. In this case, oxygen permeability of S1 was higher than S2, so this fact could have decreased the development of oxidation reactions in S2 during storage.

In general, the increases of lipid oxidation in sliced products were not very important, despite the long times of storage. Some studies have suggested that the rearing systems of pigs make them more stable to lipid oxidation reactions since pigs take natural antioxidants such as tocopherols from acorns and pasture, which are deposed into the pig tissues [16,17,37]. In line with our results, sliced Iberian chorizo did not show important increases in TBA-RS during storage of 120 days at 4 and 18 ºC, although protein oxidation significantly increased during that period [7].

### 3.4. Effect of HHP and Storage on the Sensory Analysis

Lean color of slices of S1 was not modified at day 0 and 90, but at day 180, HHP-treated slices were perceived to be darker than control (Table 7). Although after processing the appearance of slices was not modified, after long storage periods (180 days), the application of HHP could reduce the acceptability of the sliced product at the time of purchase. During storage, HHP-treated “*salchichón*” significantly increased in dark tone, while the control lean color values were not modified. In S2, lean color was affected by the storage temperature; slices stored at 20 °C were darker than those stored at 4 °C. Thus, scores for this parameter significantly increased in slices during storage at 20 °C, whereas at 4 °C, they decreased. Changes in color perceived by panelists in “*salchichón*” are difficult to compare with instrumental color, since the sensory was divided into lean and fat; meanwhile, instrumental color gives a global change in both ingredients. Despite all this, the higher darkness in S2 agrees more with the lower CIE L* values at day 180 at 20 °C than at 4 °C.

In S1, fat color was perceived to be more yellowish in HHP-treated slices at day 90. Yellow color of fat significantly increased during storage, except in the control at 4 °C. These changes are in line with the significant increase of CIE b* in HHP-treated slices stored at 4 and 20 °C. On the other hand, in S2, panelists noticed a color of fat yellower in slices stored at 20 °C at day 180 than those at refrigeration conditions. In addition, these slices stored at 20 °C showed a significant increase during storage. Results are in line with the significant increases of CIE b* during storage at 20 °C.

Odor intensity was increased in S1 during storage at 20 °C, while no significant changes were observed in this parameter in S2. In S1, odor intensity was increased during storage in HHP-treated slices at 20 °C. The storage at temperatures higher than refrigeration would have increased the development of volatile compounds from lipid oxidation, although, in general, volatile compounds formed during storage are considered negative [38].

In S1, the perception of unpleasant odors was significantly increased in HHP-treated slices during storage at both temperatures. This fact could be related to changes after HHP in proteolytic patterns and/or lipid/protein oxidation and/or microbiological changes that could have modified the original characteristics of S1. The perception of undesirable odors in “*salchichón*” would agree with increases in TBA-RS values and carbonyls content after HHP. This fact negatively affects the quality of S1, so, consequently, this type of HHP “*salchichón*” should have a shorter shelf-life of 180 days. In contrast, in S2, the perception of unpleasant odors was increased at 20 °C of storage. Lipid-derived compounds are the main volatile compounds of Iberian dry-cured meat products such as loin and ham [39]. Modifications of the original volatile profile of “*salchichón*” would enhance the perception of unpleasant odors. In fact, Rivas-Cañedo et al. [40] found that treatments of 400 MPa/10 min/12 °C increased the formation of some lipid-derived compounds and the migration of plastic bag components after HHP.

In S1, hardness intensity was significantly increased during storage in HHP-treated slices stored at 20 °C. In S2, HHP-treated slices and samples stored at 20 °C were harder than control and those stored at 4 °C at day 90. At day 180, temperature of 20 °C increased the hardness of slices. A significant increase in hardness was perceived by panelists during storage of slices at 20 °C. Increases in hardness after HHP have been also reported by Martín et al. [9] in pieces of chorizo and by Fuentes et al. [33] in sliced dry-cured ham, probably by a denaturation of proteins. Probably, the storage at temperature higher than refrigeration could also promote changes that increase the hardness of the product. Modifications of hardness were more perceivable in S2 than S1, presumably due to its lower moisture content.

In S1 and S2, juiciness was decreased in slices stored at 20 °C. Juiciness was significantly reduced during storage in control and HHP-treated “*salchichón*” at 20 °C in S2. Juiciness usually shows an opposite pattern to hardness. Reductions in juiciness in dry-cured meat product are related to low levels of fat or moisture content [41]. A lower juiciness could be explained by a higher dehydration due to storage temperature (20 °C).

Saltiness in S1 was not affected by HHP or storage conditions. However, saltiness perception was increased during storage, and significant differences were found in control stored at 4 °C. In S2 at day 0, HHP-treated “*salchichón*” were perceived as saltier than control. However, there were no significant differences between the beginning (day 0) and the end of storage (days 180), while they were perceived as less salty at day 90 in both control and HHP at 20 °C. Saltiness is associated with salt content of meat products; however, changes in the proteolytic pattern after HHP or during storage could modify the formation of peptides/amino acids with salt tastes [42].

Acid taste was not affected by treatment or storage in S1 and S2. In S1, it decreased at 90 days and increased at 180, but in S2, control products stored at 20 °C showed an acid taste more intense after storage. Initially, acid taste could be associated with the development of LAB, which acidifies the product, but changes in the proteolysis during storage could also modify the production of specific peptides or amino acids [42]. Sweet taste was not modified by HHP or storage conditions. On the other hand, the spicy taste was perceived to be more intense at the end of storage in S1. This increase is difficult to explain since this taste would be initially related to the addition of black pepper to “*salchichón*”, which would be perceived as more intense during storage.

The flavor intensity of S1 was not influenced by HHP or storage conditions; however, in S2, panelists perceived a significant increase in flavor in control at day 90 with respect to initial value. No significant changes in S2 were found at the end of storage. In general, the flavor of “*salchichón*” was well-preserved after HHP and during storage. Volatile fraction of “*salchichón*” is mainly composed of terpenes from the added spices according to specifications and the volatile compounds coming from biochemical pathways associated with microbial metabolism [40]. These volatile compounds would be well maintained in the conditions tested in this study. Similarly, cured aroma was also not affected by HHP or storage conditions; only in S2 at 20 °C was cured aroma less intense at the end of storage.

In S1, rancidity remained unchanged; however, in S2, rancidity was significantly affected by the storage temperature, with higher values at 20 °C than 4 °C. During storage, scores of HHP-treated at 4 °C and control HHP-treated at 20 °C significantly increased. Surprisingly, the increases in rancidity do not agree with TBA-RS values, but they are in line with the important increases in protein oxidation during storage in S2.

### 3.5. Principal Component Analysis

Figure 1 shows the loadings plot after the PCA. PC1 explained 43.3% of the variations of the data, whereas PC2 explained 16.4% of the variations of the data. By combining PC3, which explains 8.3%, the explanation of the variations of the data adds up to 69%. In the loading plot of panels A and B, the parameters such as microbial counts, lipid and protein oxidation, and sensory changes (appearance, odor, and rancid tastes) are in the extremes of PC1, which explains 43.3% of the variability of the results obtained. The score plot of the individuals is presented in Figure 2. PC1 allows the separation of products at day 0 (on the left) and products at day 180 (on the right). Control and HP-treated products were well grouped, so they presented similar characteristics. In S1, products stored at 4 °C at 180 days presented similar characteristics to products at 0 days, so slices at 4 °C for 180 days would present similar characteristics to those at day 0. However, in S2, refrigerated storage group was slightly separated, and these slices would present lower quality than day 0. The groups of samples stored at 20 °C were located on the right of PC1, close to oxidative reactions, rancidity, unpleasant odors, and hardness; thus, reactions were promoted at storage temperature higher than refrigeration. The storage at 20 °C would enhance the degradation of the quality of S2 more than S1. A previous study [8] on dry-cured “*salchichón*” (not sliced and not from Black category) reported that the combination of HHP treatment and storage at room temperature reached the absence of *L. monocytogenes* in 25 g in 30 days. However, this approach had the disadvantage of promoting lipid and protein oxidation during storage. Despite both products belonging to the same commercial category (Black category, according to the RD4/2014 for ham, shoulder, and loin [11]), the slight differences in the composition between both types of “*salchichón*” (higher pH and moisture content and lower polyunsaturated fatty acids content in S1 than S2), initial oxidative status (lower oxidation levels in S1 than S2), and/or differences in the plastic permeability to oxygen (higher in S1 than S2) determined this differential effect of processing and storage at room temperature in both products. Moreover, in sliced dry-cured “*salchichón*” of Black category, the importance of including other strategies to reduce oxidative phenomena at room temperature would be determinant to reach a long shelf-life of the product, such as having an adequate balance of prooxidants–antioxidants (modifying formulation) or using a protective packaging (with low oxygen permeability, or reducing light exposure) for these products, among others. If those barriers are not possible, the shortest storage times would be reached for those products at room temperature.

## 4. Conclusions

“*Salchichón*” is a traditional dry-fermented sausage generally considered a microbiologically safe product. However, processes such as the slicing or packaging can compromise its safety. Packaged sliced dry-cured sausages require long shelf-life, and they are normally stored at refrigeration temperatures. The application of hydrostatic high pressure could increase the safety of dry-cured meat products even when they are stored at room temperature. This is a demand of producers, which presents a challenge for the Iberian meat products sector. The initial characteristics of the products could affect their response to processing or storage conditions, which could also determine their shelf-life, despite having the same commercial category. Differences in their composition (moisture content and fatty acids profile), initial oxidative status, and packaging materials could influence the stability of the processed product during storage. In general, for short times of storage (less than 3 months), the sliced products had similar behavior in terms of protein oxidation or even sensory analysis; however, for long times of storage (6 months), there were slight differences for each commercial product. The differential behavior of the sliced “*salchichón*” should be taken into account to ensure an adequate shelf-life after processing.

HHP increased the food safety of products, and slight changes were detected after processing. The quality of processed products was better preserved at refrigeration than at room temperature. Storage at 20 °C, although not showing food safety problems, favored the development of color and oxidation damages. Storage at room temperature had a detrimental effect on both products (S1 and S2), but changes were more intense in S2 (higher polyunsaturated fatty acids, higher initial oxidation levels, lower moisture content) than S1. In S2, quality damages were even perceived at the sensory level, as they presented severe changes in rancidness, hardness, and the development of unpleasant odors.

In general, in the case of avoiding refrigerated storage in HHP-treated sliced Iberian “*salchichón*” or in the non-treated product, shorter shelf-life than at refrigeration conditions would be recommended. However, the improvement of packaging conditions (active packaging or reductions in oxygen permeability) could avoid quality damages in the sliced product stored at room temperature. In any case, further studies should confirm this hypothesis.

## Figures and Tables

**Figure 1 foods-11-01338-f001:**
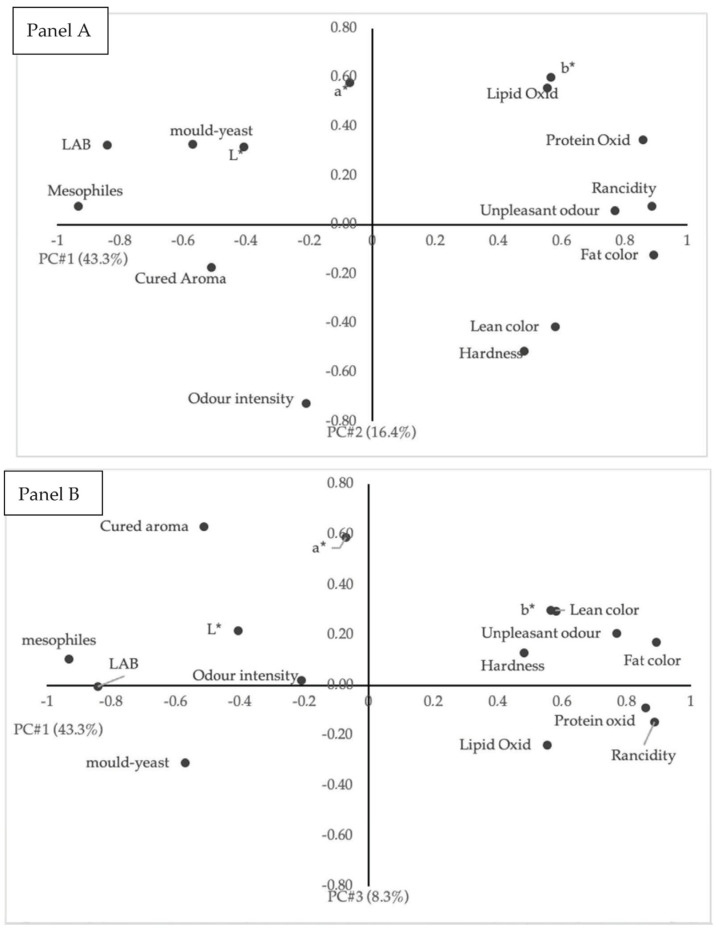
Principal component analysis. Panel (**A**): Loadings plot after principal component analysis of the variables in the plane defined by two first components (PC#1: principal component 1; PC#2: principal component 2). Panel (**B**): Loadings plot after principal component analysis of the variables in the plane defined by components (PC#1: principal component 1; PC#3: principal component 3).

**Figure 2 foods-11-01338-f002:**
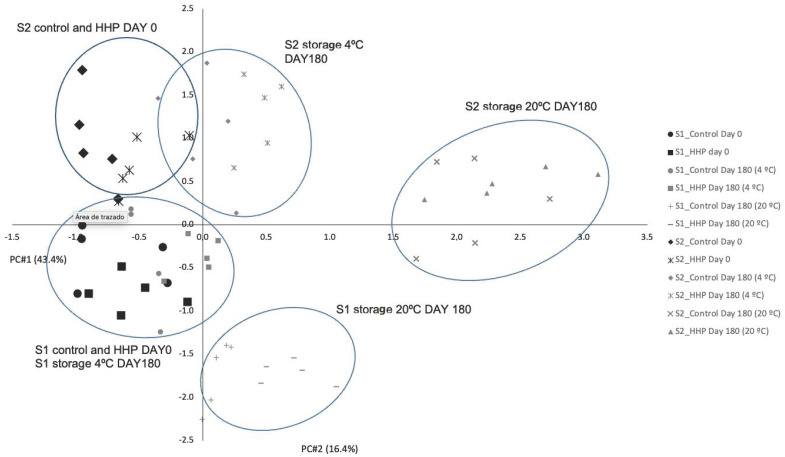
Score plot after principal component analysis of the individuals in the plane defined by two first-principal components.

**Table 1 foods-11-01338-t001:** Specifications and characteristics of “*salchichón”* from companies S1 and S2.

	S1	S2
Pig breed	100% Iberian	100% Iberian
Rearing system	Feeding in freedom with feed and supplemented with some acorn	Feeding in freedom with feed and supplemented with some acorn
Lean/fat proportion	63/37	75/25
Slice diameter	60 mm	50–55 mm
Slice thickness	≈1.6 mm	1.5 mm
Spices	Sodium chloride, black pepper, nutmeg	Sodium Chloride, black pepper
Preservatives	E250, E252	E250, E252
Other ingredients	Dextrose, corn dextrin, E331	Dextrose, glucose, Soya protein, E331, emulsifiers
Starters	None	*Lactobacillus delbrueki*
Ripening time	120 days, 75–85% RH; 7–15 °C in progressive increase	At least 120 days, 75–80% to 55–65% RH; 3–5 °C and 20–22 °C final storage.
Casing thickness	Unavailable	60–65 mm
Packaging characteristics	Plastic composite bags PET-PE 190 μm, 205 × 250 mm Oxygen permeability ≤100 cm^3^/m^2^/24 h (23 °C, 0% RH)	Plastic composite bags PA/PE 160 μm Oxygen permeability ≤ 46 cm^3^/m^2^ bar 24 h (75% RH)

**Table 2 foods-11-01338-t002:** Physical–chemical composition of sliced “*salchichón”* from companies S1 and S2.

	S1	S2	*p*-Value
pH	6.34 ± 0.21	5.60 ± 0.18	***
a_w_	0.83 ± 0.01	0.80 ± 0.00	**
%Moisture	26.31 ± 1.72	21.23 ± 1.09	**
%Protein	31.59 ± 4.40	25.81 ± 0.74	*
%Fat	27.30 ± 6.33	29.72 ± 1.98	ns

Levels of significance: ns = *p* > 0.05; * = *p* < 0.05; ** = *p* < 0.01; *** = *p* < 0.001.

**Table 3 foods-11-01338-t003:** Fatty acids profile (%) (means ± standard deviation) of sliced “*salchichón”* from companies S1 and S2.

	S1	S2	*p*-Value
C12:0	0.06 ± 0.01	0.03 ± 0.03	ns
C14:0	1.40 ± 0.03	1.19 ± 0.01	***
C16:0	24.96 ± 0.35	23.42 ± 0.19	***
C16:1	2.93 ± 0.05	2.56 ± 0.08	***
C17:0	0.26 ± 0.01	0.25 ± 0.01	ns
C17:1	0.26 ± 0.03	0.21 ± 0.01	**
C18:0	12.17 ± 0.23	11.58 ± 0.12	**
C18:1	51.25 ± 0.41	53.06 ± 0.18	***
C18:2	5.24 ± 0.25	6.20 ± 0.13	***
C18:3	0.33 ± 0.04	0.38 ± 0.01	*
C20:0	0.18 ± 0.01	0.31 ± 0.36	ns
C20:1	0.97 ± 0.03	0.79 ± 0.29	ns

Levels of significance: ns = *p* > 0.05; * = *p* < 0.05; ** = *p* < 0.01; *** = *p* < 0.001.

**Table 4 foods-11-01338-t004:** Microbiological changes (log CFU/g^−1^) after HHP of sliced “*salchichón*” samples from companies S1 and S2 during 180 days of storage.

		4 °C		20 °C		*p*-Value		
	Storage (Days)	Control	HHP	Control	HHP	T	ST	T × ST
Mesophilic aerobic counts	0	7.5 ± 0.3	7.1 ± 0.1 b	7.5 ± 0.3 a	7.1 ± 0.1 a	**	ns	ns
S1	90	7.1 ± 0.3	7.7 ± 0.1 a	6.3 ± 0.3 b	6.0 ± 0.4 b	ns	***	**
	180	7.1 ± 0.3	6.9 ± 0.2 b	6.3 ± 0.3 b	5.3 ± 0.6 c	**	***	*
P-storage		ns	***	***	***			
	0	7.8 ± 0.2 a	7.8 ± 0.1 a	7.8 ± 0.2 a	7.8 ± 0.1 a	ns	ns	ns
S2	90	7.0 ± 0.4 b	6.7 ± 0.4 b	4.8 ± 1.1 b	3.4 ± 0.7 b	*	***	ns
	180	6.7 ± 0.4 b	6.0 ± 0.4 c	3.5 ± 0.3 c	3.7 ± 0.2 b	*	***	**
P-storage			**	***	***	***		
P S1-S2	0	ns	***	ns	***			
	90	ns	**	*	***			
	180	*	**	***	***			
Lactic acid bacteria	0	7.1 ± 0.3	6.9 ± 0.1 b	7.1 ± 0.3 a	6.9 ± 0.1 a	ns	ns	ns
S1	90	6.8 ± 0.2	7.7 ± 0.1 a	5.9 ± 0.3 b	5.6 ± 0.3 b	ns	***	ns
	180	7.2 ± 0.9	6.3 ± 0.3 c	3.0 ± 0.5 c	1.6 ± 0.9 c	**	***	ns
P-storage		ns	***	***	***			
	0	7.5 ± 0.3 a	7.6 ± 0.1 a	7.5 ± 0.3 a	7.6 ± 0.1 a	ns	ns	ns
S2	90	6.8 ± 0.5 b	6.3 ± 0.5 b	5.2 ± 0.6 b	2.4 ± 0.2 b	***	**	***
	180	6.4 ± 0.3 b	5.3 ± 0.4 c	2.7 ± 0.3 c	1.2 ± 0.7 c	***	***	ns
P-storage			**	***	***			
P S1-S2	0	ns	***	ns	***			
	90	ns	***	*	***			
	180	ns	**	ns	ns			
*S. aureus*	0	2.5 ± 0.4 a	2.0 ± 0.2	2.5 ± 0.4 a	2.0 ± 0.2	**	ns	ns
S1	90	<2 b	<2	<2 b	<2	ns	ns	ns
	180	<2 b	<2	<2 b	<2	ns	ns	ns
P-storage		**	ns	**	ns			
	0	<2	<2	<2	<2	ns	ns	ns
S2	90	<2	<2	<2	<2	ns	ns	ns
	180	<2	2.0 ± 0.3	<2	<2	ns	ns	ns
P-storage		ns	ns	ns	ns			
P S1-S2	0	*	ns	*	ns			
	90	ns	ns	ns	ns			
	180	ns	ns	ns	ns			
*Cl. perfringens*	0	0.9 ± 0	<1	0.9 ± 0	<1	ns	ns	ns
S1	90	<1	1 ± 0.2	<1	1.0 ± 0.2	ns	ns	ns
	180	0.9 ± 0.1	1 ± 0.3	<1	<1	ns	ns	ns
P-storage		ns	ns	ns	ns			
	0	<1	<1	<1	<1	ns	ns	ns
S2	90	<1	<1	<1	<1	ns	ns	ns
	180	<1	<1	<1	<1	ns	ns	ns
P-storage		ns	ns	ns	ns			
P S1-S2	0	ns	ns	ns	ns			
	90	ns	ns	ns	ns			
	180	ns	ns	ns	ns			
Yeast and molds	0	2.0 ± 0.6	1.6 ± 0.4 b	2.0 ± 0.6 a	1.6 ± 0.4 a	ns	ns	ns
S1	90	1.4 ± 0.5	2.3 ± 0.4 a	1.0 ± 0.1 b	0.9 ± 0 b	ns	***	ns
	180	1.7 ± 0.6	1.1 ± 0.3 b	0.9 ± 0.1 b	0.9 ± 0.1 b	ns	**	ns
P-storage		ns	**	**	**			
	0	2.3 ± 0.7	1.3 ± 0.4	2.3 ± 0.7 a	1.3 ± 0.4 a	**	ns	**
S2	90	1.6 ± 0.7	0.9 ± 0.1	1.2 ± 0.4 b	0.9 ± 0 b	*	ns	ns
	180	1.8 ± 0.6	1.2 ± 0.4	0.9 ± 0 b	0.9 ± 0.1 b	ns	**	*
P-storage		ns	ns	**	*			
P S1-S2	0	ns	ns	ns	ns			
	90	ns	***	***	***			
	180	ns	ns	ns	ns			
Coliforms	0	3.3 ± 0.7	2.2 ± 0.5 b	3.3 ± 0.7a	2.2 ± 0.5	**	ns	ns
S1	90	3.6 ± 0.9	4.2 ± 0.5 a	2.5 ± 0.7a	3.1 ± 0.7	ns	***	ns
	180	3.3 ± 0.5	2.8 ± 0.7 b	1.2 ± 0.6 b	1.8 ± 1.2	ns	***	ns
P-storage		ns	***	***	ns			
	0	<1	1.1 ± 0.2 b	<1	1.1 ± 0.2	*	ns	*
S2	90	0.9 ± 0.1	1.8 ± 0.5 a	<1	<1	**	**	**
	180	1.2 ± 0.4	0.9 ± 0 b	1 ± 0.3	<1	ns	ns	ns
P-storage		ns	**	ns	ns			
P S1-S2	0	***	**	***	**			
	90	***	***	***	***			
	180	***	***	ns	ns			
*E. coli*	0	1.4 ± 0.2 a	1.2 ± 0.4	1.4 ± 0.2 a	1.2 ± 0.4	ns	ns	ns
S1	90	<1 b	<1	<1 b	<1	ns	ns	ns
	180	0.9 ± 0.0 b	<1	<1 b	<1	ns	ns	ns
P-storage		***	ns	***	ns			
	0	<1	<1	<1	<1	ns	ns	ns
S2	90	<1	<1	<1	<1	ns	ns	ns
	180	<1	<1	<1	<1	ns	ns	ns
P-storage		ns	ns	ns	ns			
P S1-S2	0	**	ns	**	ns			
	90	ns	ns	ns	ns			
	180	ns	ns	ns	ns			

Means ± standard deviation are represented. T: treatment effect. ST: storage temperature effect. Levels of significance: ns = *p* > 0.05; * = *p* < 0.05; ** = *p* < 0.01; *** = *p* < 0.001. a, b, c: different letters in the same column indicate significant differences during the storage (Tukey test *p* < 0.05).

**Table 5 foods-11-01338-t005:** Instrumental color changes (means ± standard deviation) after HHP of sliced “*salchichón*” samples from companies S1 and S2 during 180 days of storage.

		4 °C		20 °C		*p*-Value		
	Storage (Days)	Control	HHP	Control	HHP	T	ST	T × ST
CIE L*	0	36.7 ± 3.1	33.3 ± 2.6 b	36.7 ± 3.2	33.3 ± 2.6	*	ns	ns
S1	90	35.5 ± 3.7	37.0 ± 4.3 ab	33.6 ± 2.5	33.2 ± 1.4	ns	ns	ns
	180	36.6 ± 2.4	40.3 ± 3.1 a	31.4 ± 4.2	32.6 ± 2.5	ns	***	ns
P-storage		ns	*	ns	ns			
	0	35.8 ± 1.6	36.8 ± 2.7	35.8 ± 1.6 b	36.8 ± 2.2 b	ns	ns	ns
S2	90	36.3 ± 2.9	35.0 ± 2.7	42.1 ± 2.8 a	42.0 ± 1.2 a	ns	***	ns
	180	36.4 ± 3.0	35.1 ± 2.7	31.1 ± 3.3 c	31.7 ± 2.2 c	ns	**	ns
P-storage		ns	ns	***	***			
P S1-S2	0	ns	ns	ns	ns			
	90	ns	ns	**	***			
	180	ns	*	ns	ns			
CIE a*	0	11.8 ± 1.7	12.0 ± 1.2	11.8 ± 1.7	12.0 ± 1.2 b	ns	ns	ns
S1	90	13.0 ± 1.3	11.6 1.1	13.4 ± 2.2	14.4 ± 0.7 a	ns	*	ns
	180	14.1 ± 1.9	13.2 ± 0.9	11.4 ± 0.8	12.8 ± 1.0 b	ns	*	ns
P-storage		ns	ns	ns	****			
	0	15.9 ± 1.2	15.1 ± 2.1	15.9 ± 1.2 a	15.1 ± 2.1	ns	ns	ns
S2	90	15.2 ± 1.6	14.4 ± 1.3	14.9 ± 1.5 ab	14.7 ± 1.8	ns	ns	ns
	180	13.5 ± 2.0	14.6 ± 1.5	12.8 ± 1.5 b	13.6 ± 0.7	ns	ns	ns
P-storage		ns	ns	*	ns			
P S1-S2	0	****	***	****	***			
	90	***	****	ns	ns			
	180	ns	ns	ns	ns			
CIE b*	0	5.2 ± 0.4	4.6 ± 0.5 b	5.2 ± 0.4	4.6 ± 0.5 b	**	ns	ns
S1	90	5.5 ± 0.7	4.9 ± 0.5 b	5.3 ± 0.3	5.6± 0.5 a	ns	ns	ns
	180	5.4 ± 0.4	6.0 ± 0.4 a	5.0 ± 0.7	6.1 ± 0.9 a	**	ns	ns
P-storage		ns	**	ns	****			
	0	6.2 ± 0.7	6.8 ± 0.4	6.2 ± 0.7 b	6.8 ± 0.4 b	*	ns	ns
S2	90	6.8 ± 0.6	6.0 ± 0.5	7.8 ± 0.9 a	8.2 ± 0.7 a	ns	***	ns
	180	7.2 ± 0.7	6.8 ± 0.7	8.0 ± 1.4 a	7.7 ± 0.8 ab	ns	ns	ns
P-storage		ns	ns	***	***			
P S1-S2	0	***	*****	***	*****			
	90	***	***	*****	*****			
	180	****	ns	****	***			

Means ± standard deviation are represented. T: treatment effect. ST: storage temperature effect. Levels of significance: ns = *p* > 0.05; * = *p* < 0.05; ** = *p* < 0.01; *** = *p* < 0.001. a, b, c: different letters in the same column indicate significant differences during the storage (Tukey test *p* < 0.05).

**Table 6 foods-11-01338-t006:** Lipid (TBA-RS values, mg MDA Kg^−1^) and protein oxidation (nmols carbonyls mg protein^−1^) changes (means ± standard deviation) after HHP of sliced “*salchichón*” samples from companies S1 and S2 during 180 days of storage.

		4 °C		20 °C		*p*-Value		
	Storage (Days)	Control	HHP	Control	HHP	T	ST	T × ST
Lipid oxidation	0	0.4 ± 0.1 a	0.5 ± 0.1 b	0.4 ± 0.1 a	0.5 ± 0.1	ns	ns	ns
S1	90	0.3 ± 0.1 b	0.8 ± 0.2 a	0.3 ± 0.0 b	0.5 ± 0.0	***	*	*
	180	0.3 ± 0.1 ab	0.8 ± 0.1 a	0.3± 0.1 b	0.5 ± 0.2	***	**	*
P-storage		*	**	*	ns			
	0	1.3 ± 0.5	1.0 ± 0.1 b	1.3 ± 0.5	1.0 ± 0.1 b	ns	ns	ns
S2	90	1.4 ± 0.3	1.4 ± 0.4 b	1.5 ± 0.3	1.4 ± 0.1 b	ns	ns	ns
	180	2.0 ± 0.5	2.3 ± 0.4 a	1.5 ± 0.3	2.0 ± 0.5 a	ns	*	ns
P-storage		ns	***	ns	*****			
P S1-S2	0	****	*****	****	*****			
	90	*****	***	*****	*****			
	180	*****	*****	*****	*****			
Protein oxidation	0	4.7 ± 1.2	3.9 ± 1.2 b	4.7 ± 1.2	3.9 ± 1.2	ns	ns	ns
S1	90	4.4 ± 1.7	5.6 ± 2.7 ab	4.8 ± 1.4	5.9 ± 2.3	ns	ns	ns
	180	5.4 ± 1.1	7.8 ± 2.1 a	5.1 ± 1.6	6.1 ± 1.5	*	ns	ns
P-storage		ns	*	ns	ns			
	0	6.2 ± 0.8 b	6.3 ± 1.5 b	6.2 ± 0.8 b	6.3 ± 1.5 b	ns	ns	ns
S2	90	6.8 ± 1.9 b	7.4 ± 1.7 b	5.5 ± 2.4 b	8.2 ± 2.3 b	ns	ns	ns
	180	10.7 ± 1.7 a	10.6 ± 2.1 a	17.9 ± 3.0 a	21.7 ± 3.8 a	ns	***	ns
P-storage		****	****	*****	*****			
P S1-S2	0	***	***	***	***			
	90	ns	ns	ns	ns			
	180	***	ns	*****	*****			

Means ± standard deviation are represented. T: treatment effect. ST: storage temperature effect. Levels of significance: ns = *p* > 0.05; * = *p* < 0.05; ** = *p* < 0.01; *** = *p* < 0.001. a, b: different letters in the same column indicate significant differences during the storage (Tukey test *p* < 0.05).

**Table 7 foods-11-01338-t007:** Sensory changes (means ± standard deviation) after HHP of sliced “*salchichón*” samples from companies S1 and S2 during 180 days of storage.

		4 °C		20 °C		*p*-Value		
	Storage (Days)	Control	HHP	Control	HHP	T	ST	T × ST
Lean color	0	6.9 ± 0.6	7.2 ± 0.4 ab	6.9 ± 0.6	7.2 ± 0.4 b	ns	ns	ns
S1	90	7.2 ± 0.3	6.7 ± 0.2 b	6.8 ± 0.4	6.8 ± 0.3 b	ns	ns	ns
	180	7.1 ± 0.6	7.4 ± 0.2 a	6.9 ± 0.2	7.7 ± 0.2 a	**	ns	ns
P-storage		ns	*	ns	**			
	0	6.7 ± 0.5	7.0 ± 0.4 a	6.7 ± 0.5 b	7.0 ± 0.4 b	ns	ns	ns
S2	90	6.5 ± 0.2	6.7 ± 0.2 ab	6.9 ± 0.4 b	7.0 ± 0.4 b	ns	*	ns
	180	6.5 ± 0.3	6.3 ± 0.1 b	7.8 ± 0.2 a	7.8 ± 0.4 a	ns	***	ns
P-storage		ns	**	**	*			
P S1-S2	0	ns	ns	ns	ns			
	90	**	ns	ns	ns			
	180	ns	***	***	ns			
Fat color	0	1.5 ± 0.1	1.5 ± 0.2 b	1.5 ± 0.1 c	1.5 ± 0.2 c	ns	ns	ns
S1	90	2.0 ± 0.2	2.2 ± 0.3 a	2.0 ± 0.1 b	2.4 ± 0.3 b	*	ns	ns
	180	3.2 ± 2.3	2.5 ± 0.3 a	3.0 ± 0.2 a	3.3 ± 0.3 a	ns	ns	ns
P-storage		ns	*****	*****	*****			
	0	1.7 ±	1.8 ± 0.4	1.7 ± 0.2 c	1.8 ± 0.4 c	ns	ns	ns
S2	90	2.3 ± 0.2	2.3 ± 0.3	3.0 ± 0.4 b	3.0 ± 0.5 b	ns	ns	ns
	180	2.6 ± 0.8	2.3 ± 0.4	3.9 ± 0.7 a	4.1 ± 0.9 a	ns	**	ns
P-storage		ns	ns	***	**			
P S1-S2	0	ns	ns	ns	ns			
	90	ns	ns	***	ns			
	180	ns	ns	*	ns			
Odor intensity	0	6.2 ± 0.4	6.2 ± 0.4 a	6.2 ± 0.4 b	6.2 ± 0.4 ab	ns	ns	ns
S1	90	5.7 ± 0.5	5.6 ± 0.3 b	5.7 ± 0.3 b	5.9 ± 0.5 b	ns	ns	ns
	180	5.6 ± 2.0	6.2 ± 0.2 a	6.8 ± 0.2 a	6.7 ± 0.3 a	ns	ns	ns
P-storage		ns	***	****	***			
	0	5.7 ± 0.2	5.8 ± 0.5	5.7 ± 0.2	5.8 ± 0.5	ns	ns	ns
S2	90	6.0 ± 0.2	6.0 ± 0.4	6.0 ± 0.5	6.3 ± 0.6	ns	ns	ns
	180	6.1 ± 0.5	5.6 ± 0.3	5.8 ± 0.3	5.5 ± 0.4	ns	ns	ns
P-storage		ns	ns	ns	ns			
P S1-S2	0	*	ns	*	ns			
	90	ns	ns	ns	ns			
	180	ns	*	**	**			
Unpleasant odor	0	0.1 ± 0.1	0.0 ± 0.1 b	0.1 ± 0.1	0.0 ± 0.1 b	ns	ns	ns
S1	90	0.4 ± 0.4	0.2 ± 0.1 b	0.1 ± 0.1	0.2 ± 0.1 b	ns	ns	ns
	180	0.5 ± 0.4	0.7 ± 0.2 a	0.4 ± 0.3	0.4 ± 0.3 a	ns	ns	ns
P-storage		ns	***	ns	**			
	0	0.1 ± 0.1 b	0.3 ± 0.3	0.1 ± 0.1 b	0.3 ± 0.3 b	ns	ns	ns
S2	90	0.5 ± 0.2 a	0.5 ± 0.1	0.5 ± 0.3 ab	0.4 ± 0.2 b	ns	ns	ns
	180	0.2 ± 0.1 b	0.4 ± 0.1	1.0 ± 0.6 a	1.0 ± 0.5 a	ns	**	ns
P-storage		**	ns	**	*			
P S1-S2	0	ns	ns	ns	ns			
	90	ns	**	*	ns			
	180	ns	*	ns	ns			
Hardness	0	6.0 ± 0.7	6.1 ± 0.5	6.0 ± 0.7	6.1 ± 0.5 ab	ns	ns	ns
S1	90	6.1 ± 0.5	5.5 ± 0.6	5.4 ± 0.3	5.9 ± 0.3 b	ns	ns	*
	180	4.2 ± 2.3	6.3 ± 0.6	6.0 ± 0.4	6.7 ± 0.6 a	ns	ns	ns
P-storage		ns	ns	ns	*			
	0	5.3 ± 0.4	5.4 ± 0.3	5.3 ± 0.4 b	5.4 ± 0.3 b	ns	ns	ns
S2	90	4.8 ± 0.2	5.2 ± 0.4	5.1 ± 0.2 b	5.8 ± 0.5 b	**	*	ns
	180	5.4 ± 0.4	5.7 ± 0.5	6.1 ± 0.3 a	6.5 ± 0.4 a	ns	**	ns
P-storage		ns	ns	**	**			
P S1-S2	0	ns	*	ns	*			
	90	**	ns	ns	ns			
	180	ns	ns	ns	ns			
Juiciness	0	4.2 ± 0.6	4.3 ± 0.6	4.2 ± 0.6	4.3 ± 0.6	ns	ns	ns
S1	90	4.4 ± 0.7	4.6 ± 0.2	4.8 ± 0.3	4.4 ± 0.2	ns	ns	ns
	180	5.6 ± 1.2	4.8 ± 0.3	4.5 ± 0.3	4.0 ± 0.2	ns	*	ns
P-storage		*	ns	ns	ns			
	0	4.9 ± 0.5	4.6 ± 0.4	4.9 ± 0.5 a	4.6 ± 0.4 a	ns	ns	ns
S2	90	4.9 ± 0.3	4.7 ± 0.6	4.6 ± 0.6 a	4.2 ± 0.3 a	ns	ns	ns
	180	4.6 ± 0.3	4.2 ± 0.5	3.7 ± 0.4 b	3.3 ± 0.4 b	ns	**	ns
P-storage		ns	ns	**	***			
P S1-S2	0	ns	ns	ns	ns			
	90	ns	ns	ns	ns			
	180	ns	ns	*	*			
Saltiness	0	4.5 ± 0.2 b	4.5 ± 0.3	4.5 ± 0.2	4.5 ± 0.3	ns	ns	ns
S1	90	4.8 ± 0.2 ab	4.6 ± 0.2	4.6 ± 0.1	4.7 ± 0.2	ns	ns	ns
	180	4.9 ± 0.3 a	4.9 ± 0.3	4.8 ± 0.5	4.6 ± 0.5	ns	ns	ns
P-storage		***	ns	ns	ns			
	0	4.4 ± 0.2	4.6 ± 0.1 a	4.4 ± 0.2 a	4.6 ± 0.1 a	**	ns	ns
S2	90	4.2 ± 0.2	4.3 ± 0.2 b	4.1 ± 0.2 b	4.3 ± 0.2 b	ns	ns	ns
	180	4.5 ± 0.2	4.5 ± 0.1 ab	4.4 ± 0.1 a	4.5 ± 0.2 ab	ns	ns	ns
P-storage		ns	**	*	*			
P S1-S2	0	ns	ns	ns	ns			
	90	**	*	**	*			
	180	*	*	ns	ns			
Acid taste	0	2.5 ± 0.2 ab	2.6 ± 0.3 a	2.5 ± 0.2 a	2.6 ± 0.3 a	ns	ns	ns
S1	90	1.6 ± 0.3 b	1.5 ± 0.1 b	1.4 ± 0.1 b	1.5 ± 0.2 b	ns	ns	ns
	180	3.2 ± 1.2 a	2.9 ± 0.5 a	2.8 ± 0.4 a	2.5 ± 0.6 a	ns	ns	ns
P-storage		***	*****	*****	****			
	0	2.5 ± 0.3	2.5 ± 0.3	2.5 ± 0.3 b	2.5 ± 0.3	ns	ns	ns
S2	90	3.2 ± 0.6	2.9 ± 0.6	3.0 ± 0.5 ab	3.0 ± 0.4	ns	ns	ns
	180	2.8 ± 0.5	3.0 ± 0.3	3.2 ± 0.1 a	2.9 ± 0.6	ns	ns	ns
P-storage		ns	ns	*	ns			
P S1-S2	0	ns	ns	ns	ns			
	90	**	**	***	***			
	180	ns	ns	ns	ns			
Sweet taste	0	2.6 ± 0.1 a	2.7 ± 0.2 a	2.6 ± 0.1	2.7 ± 0.2	ns	ns	ns
S1	90	2.6 ± 0.2 a	2.7 ± 0.3 a	2.6 ± 0.3	2.7 ± 0.3	ns	ns	ns
	180	2.3 ± 0.1 b	2.1 ± 0.3 b	2.2 ± 0.4	2.2 ± 0.5	ns	ns	ns
P-storage		****	****	ns	ns			
	0	2.4 ± 0.2	2.5 ± 0.1	2.4 ± 0.2 b	2.5 ± 0.1	ns	ns	ns
S2	90	2.9 ± 0.3	2.9 ± 0.3	3.0 ± 0.4 a	2.8 ± 0.3	ns	ns	ns
	180	2.8 ± 0.4	2.8 ± 0.3	2.8 ± 0.3 ab	2.5 ± 0.2	ns	ns	ns
P-storage		ns	ns	*	ns			
P S1-S2	0	ns	ns	ns	ns			
	90	ns	ns	ns	ns			
	180	ns	*	*	ns			
Spicy taste	0	2.6 ± 0.4 ab	2.9 ± 0.4 b	2.6 ± 0.4 b	2.9 ± 0.4 a	ns	ns	ns
S1	90	1.9 ± 0.3 b	1.8 ± 0.2 c	1.9 ± 0.2 c	2.0 ± 0.6 b	ns	ns	ns
	180	3.3 ± 0.9 a	4.2 ± 0.7 a	3.7 ± 0.5 a	3.3 ± 0.3 a	ns	ns	ns
P-storage		****	*****	*****	****			
	0	2.8 ± 0.3	2.9 ± 0.3	2.8 ± 0.3 a	2.9 ± 0.3	ns	ns	ns
S2	90	2.4 ± 0.4	2.4 ± 0.2	2.3 ± 0.3 ab	2.3 ± 0.4	ns	ns	ns
	180	2.5 ± 0.2	2.8 ± 0.3	2.2 ± 0.4 b	2.5 ± 0.2	ns	ns	ns
P-storage		ns	ns	*	ns			
P S1-S2	0	ns	ns	ns	ns			
	90	*	**	*	ns			
	180	ns	*	**	**			
Flavor intensity	0	6.1 ± 0.2	6.1 ± 0.4	6.1 ± 0.2	6.1 ± 0.4	ns	ns	ns
S1	90	6.3 ± 0.3	6.4 ± 0.3	6.5 ± 0.3	6.5 ± 0.2	ns	ns	ns
	180	5.9 ± 1.2	6.7 ± 0.4	6.6 ± 0.4	6.5 ± 0.3	ns	ns	ns
P-storage		ns	ns	ns	ns			
	0	5.9 ± 0.1 b	6.1 ± 0.1	5.9 ± 0.1	6.1 ± 0.1	*	ns	ns
S2	90	6.4 ± 0.1 a	6.4 ± 0.3	6.0 ± 0.2	6.2 ± 0.3	ns	*	ns
	180	6.2 ± 0.3 ab	6.1 ± 0.4	5.9 ± 0.2	6.0 ± 0.2	ns	ns	ns
P-storage		*	ns	ns	ns			
P S1-S2	0	ns	ns	ns	ns			
	90	ns	ns	*	ns			
	180	ns	ns	*	*			
Cured aroma	0	4.7 ± 0.2 b	4.7 ± 0.7	4.7 ± 0.2 ab	4.7 ± 0.7 b	ns	ns	ns
S1	90	4.6 ± 0.3 b	4.5 ± 0.2	4.6 ± 0.3 b	4.6 ± 0.1 b	ns	ns	ns
	180	5.3 ± 0.5 a	5.4 ± 0.3	5.1 ± 0.3 a	5.4 ± 0.2 a	ns	ns	ns
P-storage		***	ns	***	***			
	0	5.1 ± 0.1 a	5.1 ± 0.4	5.1 ± 0.1 a	5.1 ± 0.4 a	ns	ns	ns
S2	90	4.6 ± 0.3 b	4.8 ± 0.3	4.6 ± 0.4 b	4.6 ± 0.5 ab	ns	ns	ns
	180	4.8 ± 0.1 ab	4.8 ± 0.3	4.1 ± 0.4 b	4.2 ± 0.3 b	ns	**	ns
P-storage		**	ns	**	*			
P S1-S2	0	**	ns	**	ns			
	90	ns	ns	ns	ns			
	180	ns	ns	**	**			
Rancidity	0	0.8 ± 0.6	0.8 ± 0.6	0.8 ± 0.6	0.8 ± 0.6	ns	ns	ns
S1	90	0.8 ± 0.3	0.5 ± 0.2	0.6 ± 0.3	1.0 ± 0.4	ns	ns	*
	180	0.7 ± 0.5	0.7 ± 0.2	1.0 ± 0.5	1.3 ± 0.8	ns	ns	ns
P-storage		ns	ns	ns	ns			
	0	0.6 ± 0.2	0.8 ± 0.3 b	0.6 ± 0.2 c	0.8 ± 0.3 b	ns	ns	ns
S2	90	1.2 ± 0.4	1.1 ± 0.2 ab	1.4 ± 0.6 b	1.2 ± 0.5 b	ns	ns	ns
	180	0.9 ± 0.3	1.5 ± 0.4 a	2.4 ± 0.5 a	2.7 ± 0.6 a	ns	***	ns
P-storage		ns	*	***	***			
P S1-S2	0	ns	ns	ns	ns			
	90	ns	**	*	ns			
	180	ns	*	**	*			

Means ± standard deviation are represented. T: treatment effect. ST: storage temperature effect. Levels of significance: ns = *p* > 0.05: * = *p* < 0.05: ** = *p* < 0.01: *** = *p* < 0.001. a, b, c: different letters in the same column indicate significant differences during the storage (Tukey test *p* < 0.05).

## Data Availability

The authors confirm that the data supporting the findings of this study are available within the article and the raw data that support the findings are available from the corresponding author, upon reasonable request.

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
