# Peer review of "Effect of High-Hydrostatic-Pressure Processing and Storage Temperature on Sliced Iberian Dry-Cured Sausage (“Salchichón”) from Pigs Reared in Montanera System"

_foods, 2022, doi:10.3390/foods11091338_

Round 1

Reviewer 1 Report

The article “Effect of high hydrostatic pressure processing and storage tem perature on sliced Iberian dry-cured sausage (“salchichón”) from pigs reared in Montanera system” is well written.

The comments are as follows:

Comment 1: Why this product is important for consumers and what is recent data on its consumption.

Comment 2: Line 32-33: Does these condition effect the quality of meat? If yes, then elaborate the changes under these conditions?

Comment 3: There are many studies available on this fermentation? What is novelty of this study?

Comment 4: How this study can change the consumer acceptability besides consumers are tilt the normal products available in the market.

Comment 5: Also briefly explain the industrial and practical significance of this study. 

Comment 6: Line 87: why the samples were stored for 180 days.? Is it standardized?

Comment 7: Line 96: why the 600 MPa was used. Meanwhile, in earlier articles the use of HHP is different on meat products.

Comment 8: If possible, add the Ca-ATPase and surface hydrophobicity to support the carbonyl contents.

Comment 9: Need to add more literature that can back the fatty acids composition.

Comment 10: Section 3.3: As per my knowledge, lipid and protein oxidation directly associated with each other, but no enough discussion was found here

Comment 11: Does the HPP decreased the appearance of products? Processing process could effect? What and why these changes were induced?

Comment 12: Add the practical significance of this study in conclusion section.

Author Response

REVIEWER 1:

The article “Effect of high hydrostatic pressure processing and storage temperature on sliced Iberian dry-cured sausage (“salchichón”) from pigs reared in Montanera system” is well written.

The comments are as follows:

Comment 1: Why this product is important for consumers and what is recent data on its consumption.

Response 1: In 2020 (the most recent data available), the consumption of meat products (fermented, dry-cured and cooked) increased respect to the previous year. Concretely, the consumption of “salchichón” per capita in 2020 was of 440g. We have added this information in the manuscript (lines 39-42):

In 2020, the consumption of meat products in Spain (fermented, dry-cured and cooked) increased compared to the previous year. The “salchichón” consumption represented the 10% of the total [4].

Comment 2: Line 32-33: Does these condition effect the quality of meat? If yes, then elaborate the changes under these conditions?

Response 2: This information has been added in the manuscript (lines 35-36):

Feeding system determines the stability of sliced Iberian dry-cured meat products during storage [2].

Comment 3: There are many studies available on this fermentation? What is novelty of this study?

Response 3: This paper does not evaluate the fermentation process; the objective is to evaluate post-packaging changes associated to hydrostatic high pressure and the temperature of storage. This information has been added in the manuscript. In addition, the novelty of the study has also been included (Lines 75-77):

Consequently, the aim of this work has been to evaluate post-packaging changes associated to hydrostatic high pressure and the temperature of storage on two types of sliced Iberian “salchichón” with different composition.

Comment 4: How this study can change the consumer acceptability besides consumers are tilt the normal products available in the market.

Response 4: The application of hydrostatic high pressure should not affect the acceptability of the product since the consumers do not receive that information. However, sometimes some producers have safety problems in the sliced product and they apply this treatment to fulfil legislation requirements. Industrial do not want to reduce the original quality of the product after processing and they would like to maintain the original quality of the dry-cured sausage.

Comment 5: Also briefly explain the industrial and practical significance of this study. 

Response 5: “Salchichón” is a traditional sausage, produced mainly by artisanal practices, which is generally considered a microbiologically safe product. However, processes such as the slicing or packaging, can compromise their safety.  Packaged sliced dry-cured sausages require long shelf-life, and they are normally stored at refrigeration temperatures. Producers also demand a possible storage without refrigeration for these products, which supposes a challenge for Iberian meat products sector. The application of hydrostatic high pressure could increase the safety of dry-cured meat products even when they are stored at room temperature. The initial characteristics of the products could affect their response to processing or storage conditions which could also determine their shelf-life. This information has been briefly added in the manuscript (lines 49-57):

Salchichón” is generally considered a microbiologically safe product. However, processes such as the slicing or packaging, can compromise their safety. Packaged sliced dry-cured sausages require long shelf-life, and they are normally stored at refrigeration temperatures. Producers also demand a possible storage without refrigeration of the sliced product, which supposes a challenge for Iberian meat products sector. Microbiological spoilage and lipid oxidation reactions in food are affected by temperature, both are enhanced with increasing temperature affecting final quality. Therefore, shelf life of “salchichón” is influenced, among other factors, by both the storage temperature and the presentation format [7-9].

Comment 6: Line 87: why the samples were stored for 180 days.? Is it standardized?

Response 6: The date of preference consumption is generally 6-8 months but it varies between companies. We decided that 180 days was enough time to observe the effect of processing and storage temperature. This information has been added (lines 106-108):

Storage time was decided according to the date of preference consumption of these products, which is generally less than 6-8 months.

Comment 7: Line 96: why the 600 MPa was used. Meanwhile, in earlier articles the use of HHP is different on meat products.

Response 7: Generally, meat products companies’ are applying treatment conditions of 600 MPa for 6-8 minutes to inactivate Listeria monocytogenes. Pressure intensity of 600 MPa is the maximum intensity of pressure reached by commercial equipments. Times between 6-8 minutes are sufficient for the inactivation of Listeria since longer times of processing does not ensure higher inactivation of this pathogen. In addition, long times of processing reduces the cost effectiveness of the high pressure equipments, so generally longer times than 10 minutes are not being applied at commercial level because that reduces their profitability.

The earlier studies about the application of HHP tested lower intensities of pressure in dry-cured meat products because the objective initially was to know the effect of pressure intensity on the quality of the product. However, later as the main problem of dry-cured meat products was slicing and the contamination by Listeria, the most recent studies were focused in the inactivation of this pathogen. These studies established that treatments of 400MPa were not sufficient for the inactivation of Listeria in dry-cured meat products.

Comment 8: If possible, add the Ca-ATPase and surface hydrophobicity to support the carbonyl contents.

Response 8: We are very sorry but at this moment it is impossible for us to add the analysis of the Ca-ATPase and surface hydrophobicity of the meat product, however we consider them very interesting and they are quite unknown in dry-cured meat products. So, we will try to add this type of analysis in futures studies because maybe they clarify the effect of processing and storage on protein oxidation.

Comment 9: Need to add more literature that can back the fatty acids composition.

Response 9: More discussion about the effect of fatty acids profile in meat stability has been added (lines 208-209).

Fatty acids profile could affect the stability of dry-cured meat products after processing or storage, since unsaturated fatty acids are easily oxidized [18].

Comment 10: Section 3.3: As per my knowledge, lipid and protein oxidation directly associated with each other, but no enough discussion was found here

Response 10: More discussion about this has been added (lines 422-427).

Fuentes reported that pre-sliced dry-cured ham was more susceptible to oxidative reactions after HHP and subsequent refrigerated storage, since paired or cross-linked reactions between lipid and protein oxidation may have occurred. However, the literature provides conflicting results as not always positive correlations between protein and lipid oxidation are found. In fact, the protein oxidation after HHP is a very recent topic [36].

Comment 11: Does the HPP decreased the appearance of products? Processing process could effect? What and why these changes were induced?

Response 11: The application of HHP should not affect the appearance of the product, but HHP treatments are associated with an increase in lipid oxidation, so storage conditions that could favour these reactions, such as long storage periods, have been established, as well as comparing standard refrigerated storage with storage at room temperature. In any case, a panel of tasting judges has been used to evaluate if there are any changes in organoleptic quality after treatments and storage. We have added some discussion about this (line 444-446):

Although after processing the appearance of slices was not modified, after long storage periods (180 days) the application of HHP could reduce the acceptability of the sliced product at the time of purchase.

Comment 12: Add the practical significance of this study in conclusion section.

Response 12: Practical application of this study was added to conclusions section (lines 578-586).

“Salchichón” is a traditional dry-fermented sausage generally considered a microbiologically safe product. However, processes such as the slicing or packaging, can compromise their safety.  Packaged sliced dry-cured sausages require long shelf-life, and they are normally stored at refrigeration temperatures. The application of hydrostatic high pressure could increase the safety of dry-cured meat products even when they are stored at room temperature. This is a demand of producers, which supposes a challenge for Iberian meat products sector. The initial characteristics of the products could affect their response to processing or storage conditions which could also determine their shelf-life, despite they had the same commercial category.

Reviewer 2 Report

Comments to authors:

The authors present the effect of HHP and storage temperature on salchichón from pigs reared in the Montanera system. Please describe briefly and clearly the novelty and importance of this research as there are several latest articles on the same topic, i.e.,

  • High-pressure processing and storage temperature on Listeria monocytogenes, microbial counts and oxidative changes of two traditional dry-cured meat products. https://doi.org/10.1016/j.meatsci.2020.108273
  • Effect of high hydrostatic pressure processing and storage temperature on food safety, microbial counts, colour and oxidative changes of a traditional dry-cured sausage.

https://doi.org/10.1016/j.lwt.2020.109462

  • Colour modification in a cured meat model dried by Quick-Dry-Slice process® and high pressure processed as a function of NaCl, KCl, K-lactate and water contents.

https://doi.org/10.1016/j.ifset.2011.09.005

  • The effects of high-pressure treatment and of storage periods on the quality of vacuum-packed “salchichón” made of raw material enriched in monounsaturated and polyunsaturated fatty acids.

https://doi.org/10.1016/j.ifset.2006.09.005

Please carefully make the corrections suggested in the text. 

Author Response

The authors present the effect of HHP and storage temperature on “salchichón” from pigs reared in the Montanera system. Please describe briefly and clearly the novelty and importance of this research as there are several latest articles on the same topic, i.e.,

  • High-pressure processing and storage temperature on Listeria monocytogenes, microbial counts and oxidative changes of two traditional dry-cured meat products. https://doi.org/10.1016/j.meatsci.2020.108273
  • Effect of high hydrostatic pressure processing and storage temperature on food safety, microbial counts, colour and oxidative changes of a traditional dry-cured sausage.

https://doi.org/10.1016/j.lwt.2020.109462

  • Colour modification in a cured meat model dried by Quick-Dry-Slice process® and high pressure processed as a function of NaCl, KCl, K-lactate and water contents.

https://doi.org/10.1016/j.ifset.2011.09.005

  • The effects of high-pressure treatment and of storage periods on the quality of vacuum-packed “salchichón” made of raw material enriched in monounsaturated and polyunsaturated fatty acids.

https://doi.org/10.1016/j.ifset.2006.09.005

Please carefully make the corrections suggested in the text. 

Response:

A briefly text indicating the importance of this research has been added (lines 60-64; 72-74).

Industrials are applying high pressure treatments to increase the safety of these products due to pre-packaging manipulation. Sometimes some undesirable quality damages may appear. This is especially important for top-quality Iberian meat products (from 100% Iberian pigs reared outdoors). No previous study has evaluated the effect of HHP and storage conditions on sliced top quality “salchichón”. In addition, two types of products manufactured in different companies were studied to obtain more representative results.

Reviewer 3 Report

The manuscript received for review investigates the effects of high hydrostatic pressure and storage time on two different storage temperatures of sliced Iberian dry-cured sausage (“salchichón”) on numerous quality parameters. Selection of tested quality responses indicates on systematic approach to the overall technological evaluation.

Title of the manuscript is precise, informative and it is clearly and concisely written.

Introduction section is elaborative with sufficient and contemporary literature data.

The Materials and Methods section describes in enough detain all conducted analysis of technological quality testing

The results and discussion section is appropriate, although some supplementation of discussion is needed.

Further PCA discussion and changes to the figures are needed.  

Conclusion section is appropriate and concludes all investigations conducted in this reserach.

All detail comments and some needed minor corrections are noted in manuscripts’ pdf file.

Reviewer recommendation: Minor revision.

Author Response

The manuscript received for review investigates the effects of high hydrostatic pressure and storage time on two different storage temperatures of sliced Iberian dry-cured sausage (“salchichón”) on numerous quality parameters. Selection of tested quality responses indicates on systematic approach to the overall technological evaluation.

Comment 1: Title of the manuscript is precise, informative and it is clearly and concisely written.

Response 1: Thank you for your comment.

Comment 2: Introduction section is elaborative with sufficient and contemporary literature data.

The Materials and Methods section describes in enough detain all conducted analysis of technological quality testing

The results and discussion section is appropriate, although some supplementation of discussion is needed.

Response 2: Thank you for your comments.

Comment 3: Further PCA discussion and changes to the figures are needed.  

Response 3: We have added more discussion to the PCA section (lines 538-548; 552-556; 562-568).

By combining PC3, which explains 8.3%, the explanation of the variations of the data adds up to 69%. In the loading plot of panels A and B, the parameters such as microbial counts, lipid and protein oxidation and sensory changes (appearance, odour and rancid tastes) are in the extremes of PC1 which explains the 43.3% of the variability of the results obtained.

A previous study [8] in dry cured “salchichón” (not sliced and not from Black category) reported that the combination of HHP treatment and storage at room temperature reached the absence of L. monocytogenes in 25-g in 30 days. However, this approach had the disadvantage of promoting lipid and protein oxidation during storage.

Moreover, in sliced dry-cured “salchichón” with Black category, the importance of in-cluding other strategies to reduce oxidative phenomena at room temperature would be determinant to reach a long shelf-life of the product such as having an adequate balance of prooxidants-antioxidants (modifying formulation), using a protective packaging (with low oxygen permeability, or reducing light exposure) for these products, among others. If those barriers are not possible, shortest storage times would be reached for those products at room temperature.

Comment 4: Conclusion section is appropriate and concludes all investigations conducted in this research.

Response 5: Thank you for your comment.

Comment 6: All detail comments and some needed minor corrections are noted in manuscripts’ pdf file.

Response 6: Thank you for your corrections, we have included them in the manuscript.
